

# Deregulation mechanisms and therapeutic opportunities of p53-responsive microRNAs in diffuse large B-cell lymphoma

Elena N. Voropaeva[1,2], Yuriy L. Orlov[3,4], Anastasia B. Loginova[2], Olga B. Seregina[2], Vladimir N. Maksimov[1,2] and Tatiana I. Pospelova[2]

[1] Research Institute of Internal and Preventive Medicine - Branch of the Federal State Budget Scientific Institution "The Federal Research Center Institute of Cytology and Genetics of Siberian Branch of the Russian Academy of Sciences", Novosibirsk, Russia
[2] Novosibirsk State Medical University of the Ministry of Health of the Russian Federation, Novosibirsk, Russia
[3] The Digital Health Center, I.M Sechenov First Moscow State Medical University, Moscow, Russia
[4] Agrarian and Technological Institute, Patrice Lumumba Peoples' Friendship University of Russia, Moscow, Russia

## ABSTRACT

Here, we have discussed the molecular mechanisms of p53-responsive microRNAs dysregulation in response to genotoxic stress in diffuse large B-cell lymphoma (DLBCL) patients. The role of micro ribonucleic acids (microRNAs) in p53-signaling cellular stress has been studied. MicroRNAs are the small non-coding RNAs, which regulate genes expression at post-transcriptional level. Many of them play a crucial role in carcinogenesis and may act as oncogenes or suppressor of tumor growth. The understanding of the effect of p53-responsive microRNA dysregulation on oncogenesis achieved in recent decades opens wide opportunities for the diagnosis, prediction and of microRNA-based cancer therapy. Development of new bioinformatics tools and databases for microRNA supports DLBCL research. We overview the studies on the role of miRNAs in regulating gene expression associated with tumorigenesis processes, with particular emphasis on their role as tumor growth-suppressing factors. The starting point is a brief description of the classical microRNA biogenesis pathway and the role of p53 in regulating the expression of these molecules. We analyze various molecular mechanisms leading to this dysregulation, including mutations in the *TP53* gene, DNA methylation, changes in host-genes expression or microRNA gene copy number, mutations in microRNA and microRNA biogenesis genes.

Corresponding authors
Elena N. Voropaeva, vena.81@mail.ru
Yuriy L. Orlov, orlov@bionet.nsc.ru

## INTRODUCTION

The *TP53* gene plays a central role in limiting tumor development. The p53 protein it encodes is involved cell cycle arrest and DNA repair to protect genome integrity or irreversible activation of apoptosis/aging programs. This ensures the removal of irreparably damaged and malignant cells and made possible by activating or suppressing transcription

of several clusters of hundreds of functionally related genes having p53 response elements (*Capaccia et al., 2022*).

In recent decades, the key role of micro ribonucleic acids (microRNAs) in p53-signaling cellular stress has been established. MicroRNAs are the small non-coding RNAs, which regulate genes expression at post-transcriptional level. Many of them play a crucial role in carcinogenesis and may act as oncogenes or suppressor of tumor growth. The role of microRNAs in tumors development has been actively studied in recent decades, because they participate in all cell processes that modulate malignant cell potential, such as cell cycle control, DNA damage response, DNA differentiation, proliferation, apoptosis, adhesion, metabolic reprogramming, epithelial-mesenchimal transition, *etc.* (*Ghosh et al., 2021*).

The efforts of a numerous researchers have made it possible to identify a large set of p53-responsive microRNAs (*Ghatak et al., 2021*; *Madrigal et al., 2021*; *Grespi et al., 2016*; *Smolarz et al., 2022*). One or the other of them is down-regulated in different cancers relative to appropriate normal tissue, further supporting the idea that they might be part of a tumor-suppressing program (*Voropaeva et al., 2020*; *La et al., 2018*; *Braun et al., 2008*). In recent review the authors have mentioned the importance of p53-microRNA network in diagnostic and therapeutic approaches for numerous cancers (*Sargolzaei, Etemadi & Alyasin, 2020*).

Understanding the effect of p53-responsive microRNA dysregulation on oncogenesis achieved in recent decades opens up wide opportunities for the diagnosis, prediction and of microRNA-based cancer therapy. The latter seems to be the most difficult. Despite the encouraging results of a growing number of studies indicating the potential of microRNAs as therapeutic agents or therapy targets, there are a number of problems that need to be resolved (*Winkle et al., 2021a*; *Seo et al., 2019*; *Ashrafizadeh et al., 2021*; *Hashemi & Gorji-Bahri, 2020*; *Orlov et al., 2023*), one of which is as follows. The hypothesis of the usefulness of using microRNAs as targets of therapy is largely based on *in vitro* experiments with cultures of tumor cell lines. In these models took into account of one reason or another reason of the studied microRNAs dysregulation (copy number alteration, nucleotide sequence changes, epigenetic modifications, *etc.*) (*Peng et al., 2020*; *Chim et al., 2011a*; *Xu et al., 2022*; *Fatema, Larson & Barrott, 2022*). In experiments the elimination of this cause leads to the restoration of normal microRNA expression, sensitivity to chemotherapy and apoptosis neoplastic cells (*Hedström et al., 2013*; *Larrabeiti-Etxebarria et al., 2019*). At the same time, various aberrations may be the basis for microRNA expression deregulation in the same type of malignant neoplasm. The existing inter-patient and intra-tumor heterogeneity will inevitably prevent the transfer of success achieved in preclinical conditions to real patients.

In this review, important issues concerning the molecular mechanisms of interested microRNA dysregulation in response to genotoxic stress in diffuse large B-cell lymphoma (DLBCL) patients are discussed. We provide an overview of previous studies on the role of miRNAs in regulating gene expression associated with tumorigenesis processes, with particular emphasis on their role as tumor growth-suppressing factors. The starting point is a brief description of the classical microRNA biogenesis pathway and the role of p53 in regulating the expression of these molecules. Subsequently, the results of studies showing dysregulation of specific associated with the response to p53 microRNAs in DLBCL

are discussed. We analyze various molecular mechanisms leading to this dysregulation, including mutations in the *TP53* gene, DNA methylation, changes in host-genes expression or microRNA gene copy number, mutations in microRNA and microRNA biogenesis genes.

This cross-disciplinary review is intended for a wide range of readers, including researchers and physicians in the fields of hematology, oncology, laboratory genetics, and medicine.

## METHODOLOGY SECTION

A search was conducted for publications in databases PubMed, Web of Science, Scopus with the use of keywords and word combinations: DLBCL; p53-responsive microRNAs; *TP53* gene; miR-34; miR-129; miR-203; miR-143; mir-145; biogenesis; expression; methylation; mutations; deletion. We have analyzed publications issued from 2010 to 2024, limited our search to studies conducted in humans and human cell lines and published in English. A small number of earlier publications, including those of historical interest, have been included in this review.

MicroRNA-target interactions for *Homo sapiens* were described according to miRTarBase, which contains only experimentally validated data (*Huang et al., 2022*). The data of the DLBCL standard karyotyping were obtained from the Mitelman Database of Chromosome Aberrations and Gene Fusions in Cancer (https://mitelmandatabase.isb-cgc.org/) (*Mitelman Database of Chromosome Aberrations and Gene Fusions in Cancer, 2024*). Cases of DLBCL with monosomy by chromosomes 1, 5, 7, 11, 14 or deletion of 1p36, 5q32, 7q32, 11q23, 11p11, 14q32 loci were selected. The Affymetrix SNP 6.0 microarrays and NGS data were obtained from the cBioPortal for Cancer Genomics database (https://www.cbioportal.org/) (*Gao et al., 2013*). Chromosomal copy-number variation in the genome loci chr11:111,383,663-111,384,240 (*MIR-34B/C* gene), chr7:127,847,925-127,847,996 (*MIR-129-1* gene), chr11:43,602,944-43,603,033 (*MIR-129-2* gene), chr14:104,583,742-104,583,851 (*MIR-203* gene), chr5:148,808,481-148,810,296 (*MIR-143* and *MIR-145* gene) according to GRCh37/hg19 assembly were analyzed.

A notion of the combined detection of the studied genes methylation was obtained using the OncoPrinter tool (https://www.cbioportal.org/oncoprinter) (*Gao et al., 2013*).

### Micro-RNAs biogenesis

Mature microRNAs are synthesized during multistage biogenesis. The canonical pathway of this process includes the stages of the microRNA gene transcription by RNA polymerase and formation of the primary microRNA transcript (pri-miR); cleavage of pri-miR in the nucleus by nuclear RNase III Drosha and its cofactor DGCR8/Pasha to a microRNA precursor (pre-miR), which is transported from the nucleus to the cell cytoplasm using the exportin protein; cleavage of pre-miR in the cytoplasm using another RNase III Dicer to a microRNA duplex consisting of mature and antisense microRNA (miR). One of the duplex strands is subsequently included in the RNA-induced silencing complex (RISC) by the Argonaute protein and serves to suppress the corresponding targets by repressing translation or by directing mRNA degradation (*Gulyaeva & Kushlinskiy, 2016*).

There are many reasons for the disruption of normal microRNA expression, which are based on both genetic and epigenetic changes. All of them lead to disruption of the processes of transcription and subsequent microRNA processing (*Zhang, Liao & Tang, 2019*).

## Description of p53-responsive microRNAs

The p53-mediated cellular response to genotoxic stress is carried out due to changes in the expression spectrum of microRNAs, namely the activation of tumor suppressor microRNAs (miR-15a/16, miR-23a, miR-29, miR-107, miR-143, miR-145, miR-192, miR-194, miR-215, miR-605, let-7 and other) and suppression of oncogenic microRNAs (miR-17-92 cluster, miR-221/2222 and other), which is possible to implement both directly through transcription-dependent mechanisms and indirectly (*Capaccia et al., 2022*; *Zhang, Liao & Tang, 2019*; *Jacques et al., 2020*; *Kaller et al., 2022*).

A number of studies have shown a decrease in the level of tumor suppressor microRNAs miR-34a, miR-34b, miR-34c, miR-129, miR-203, miR-145, miR-143 in DLBCL compared with non-tumor lymphoid tissue (*Hedström et al., 2013*; *Akao et al., 2007*; *Isaadi et al., 2021*; *Yamagishi et al., 2015*; *Zheng et al., 2021*; *Leivonen et al., 2017*; *Larrabeiti-Etxebarria et al., 2023*). An increase in the expression of miR-145 and miR-143 in DLBCL tissue compared to healthy B-lymphocytes has been recorded only in the study by *Lawrie et al. (2009)*.

The listed microRNAs are p53-responsive. It is important to note that some of them are involved in a feedback loop complex with p53, which contributes to enhanced, better controlled and fine-tuning of the cellular response to genotoxic stress (*Cabrita et al., 2016*). For example, miR34a may create a positive feedback loop with p53 by inhibiting the deacetylase SIRT1, which lead to p53 acetylation and activation (*Gong et al., 2023*; *Ong & Ramasamy, 2018*), whereas miR-129 and miR-145 directly target the anti-apoptotic factors MDM2 and MDM4, and, thus, counteract the MDM2 and MDM4-mediated suppression of p53 signaling (*Yao et al., 2021*; *Zeinali et al., 2019*).

According to miRTarBase, which contains data on microRNA-target interactions validated experimentally by reporter assay, western blot, microarray and next-generation sequencing experiments, p53-responsive microRNAs, such as microRNAs miR-34 family, miR-143/145 cluster, miR-129 and miR-203 have many common targets, namely pro-oncogenic transcription factors, positive regulators of the cell cycle at the G1/S transition checkpoint, anti-apoptotic factors, neoangiogenesis factors, participants in oncogenic signaling pathways, as well as DNA methyltransferases, which are significant in lymphomagenesis and oncogenesis in general (Table 1) (*Larrabeiti-Etxebarria et al., 2019*).

The *MIR-34A* gene is mapped on 1p36.22. The *MIR-34B/C* gene is mapped on 11q23.1 and is responsible for the sequence of the bicistronic transcript, from which mature microRNAs miR-34b and miR-34c are subsequently formed. *MIR-34A* and *MIR-34B/C* are located in the host genes of long non-coding RNAs, such as EF570048 and BC021736, respectively (*Asmar et al., 2014*).

MicroRNAs of the miR-34 family are characterized by a high degree of homology in the seed sequence, which is necessary for binding to mRNA targets. Using bioinformatics and
**Table 1** Selected significant in lymphomogenesis oncogenic targets, according to miRTarBase (*Ong & Ramasamy, 2018*).

| MicroRNA | MicroRNA targets |
| --- | --- |
| miR-34a | MYC, CDK4/6, BCL2, NOTCH1/2, VEGFA, MAP3K9, SOX2 |
| miR-34b | MYC, CDK4/6, BCL2, NOTCH1/2, VEGFA, MAP2K1, SOX2 |
| miR-34c | MYC, CDK4/6, BCL2, MDM4, NOTCH1/4, PDGFRA/B, SOX2 |
| miR-129 | MYC, CDK6, NOTCH1, BCL2L2, MAPK6, MAP3K2, PDGFRA, SOX4 |
| miR-143 | BCL2, MDM2, KLF4, DNMT3A, MAPK1/7, MAP3K, PDGFRA, SOX2 |
| miR-145 | MYC, CDK4/6, CDKN1A, STAT1, KLF4/5, MAP3K3, MAP2K4, MAP3K11, VEGFA, MDM2, SOX2/9/11 |

experimental studies, microRNAs of the miR-34 family have been shown to have a large number of common targets (*Zhang, Liao & Tang, 2019*).

Human microRNA miR-129 is encoded by *MIR-129-1* and *MIR-129-2*, which located intergenic on 7q32.1 and 11p11.2, respectively (*Yu et al., 2013*). The targets of miR-129 are mRNAs of several oncogenes, the most studied of which are CDK6, PDGFRA, FOXP1, HMGB1 and the stem cell transcription factors SOX4, EIF2C3 and CAMTA1 (*Yu et al., 2013*; *Wang, Luo & Guo, 2014*; *Voropaeva et al., 2022a*).

The *MIR-143* and *MIR-145* genes are located 1.7 kb apart on 5q32. The host-gene of this microRNA cluster is the *CARMN* non-coding RNA gene. It is assumed that miR-143 and miR-145 are formed from a common precursor and their functions are co-operated (*Pidíkova, Reis & Herichova, 2020*). It has been shown that the levels of miR-143 and miR-145 increase in response to stress through the PI3K/Akt and p53-mediated pathways. The targets of miR-143 are such well-known oncogenes as SOX2, KLF4, DNMT3A, MAPK1/7, MAP3K, BCL2, MDM2 and PDGFRA. miR-145 is involved in the post-transcriptional regulation of SOX2/9/11, STAT1, CDKN1A, KLF4/5, MAP3K3, MAP2K4, MAP3K11, MYC, CDK4/6, VEGFA, MDM2, *etc.* (*Voropaeva et al., 2023d*; *Zhang et al., 2022*).

The *MIR-203* gene is located in the intergenic region of the chromosomal locus 14q32.33. Studies show that the microRNA miR-203 encoded by this gene, besides targeting CDK6, SOX2/4, VEGFA, MYD88, MDM4, BCL2L2, is also involved in post-transcriptional control of PI3K/AKT, SRC, RAS/MAPK and JAK/STAT3 signaling pathways participants. Its predicted targets are proto-oncogenes such as DNMT3B and CREB1 (*Voropaeva et al., 2022a*).

## Impaired p53 function as a cause of deregulation of p53-responsive microRNAs

Mutations of the *TP53* gene are the commonest molecular genetic changes in malignant neoplasms including DLBCL. A recent review has addressed the importance of *TP53* gene mutations in DLBCL that is relevant to poor prognosis. The better understanding of abnormalities of p53 is meant for the basis of development of better therapeutic strategy for DLBCL (*Wen et al., 2024*). Most of them affect the DNA-binding domain (Fig. 1A) and can be divided into three types based on their effect: loss-of-function, gain-of-function and dominant-negative. It is critically important because by its function the p53 protein is a

transcription factor that is involved in the expression of numerous protein and microRNA-coding genes (*Wen et al., 2024*). Over the years, several published research strongly established the gain of function mutant p53 promoting aberrant expression of several miRNAs leading to cells reprogramming and pluripotency, provoke chemoresistance and cell survival to drive different cancer phenotypes (*Ghatak et al., 2021*; *Madrigal et al., 2021*). On the other hand, the loss-of-function mutations lead to disruption of the ability of p53 to contact regulatory elements of canonical target microRNA genes (*Strano et al., 2007*). A predominant repression effect on miRNA expression was found for p53 mutants (*Madrigal et al., 2021*; *Zhang et al., 2016*).

Also, both wild type and mutant p53 may affect microRNA processing. Wild-type p53 has been shown to regulate microRNA maturation by facilitating Drosha-mediated processing of pri-miRNA into pre-miRNA (*Liu et al., 2017*) whereas mutant p53 may break biogenesis and decrease production of mature microRNAs not only by inhibiting the interaction between Drosha and pri-microRNA (*Yao et al., 2021*; *Asmar et al., 2014*; *Speciale et al., 2020*; *Navarro & Lieberman, 2015*), but also by inhibiting p63 and reducing Dicer expression (*Liu et al., 2017*).

*MIR-34A, MIR-34B/C, MIR-129-1, MIR-129-2, MIR-143* and *MIR-145* genes, which contain the sequence of the corresponding microRNAs, are p53-responsive. It has been proven that the promoters of all of these genes except *MIR-203* contain p53-responsive elements triggering transcription by binding of 53 protein (*Asmar et al., 2014*; *Poli, Seclì & Avalle, 2020*; *Rihani et al., 2015*; *Madrigal et al., 2021*). The participation of p53 in the regulation of expression at the post-transcriptional level has been described for microRNAs miR-203 and the miR-143/145 cluster (*Larrabeiti-Etxebarria et al., 2023*; *Yao et al., 2021*; *Asmar et al., 2014*; *Davis-Dusenbery & Hata, 2010*; *Bruijn et al., 2023*).

The decrease in the expression of microRNAs miR-34a, miR-34b, miR-34c, miR-129, miR-203, miR-145, miR-143 in tumors with mutant p53 status has been shown (*Navarro & Lieberman, 2015*; *Poli, Seclì & Avalle, 2020*; *Fang et al., 2023*). Unfortunately, there is no large-scale current study of the expression repertoire of p53-responsive microRNAs depending on the mutational status of *TP53* in tumor samples from patients with DLBCL.

However, mutations in the *TP53* gene are verified in 20% of cases of DLBCL at the time of diagnosis, with an even higher frequency in relapsed lymphoma (*Li et al., 2023*).

Considering the mentioned above, impairment of p53 function as a result of mutations could potentially be one of the significant reasons for the decrease in the level of p53-responsive microRNAs of the miR-34, miR-203, miR-129 family and the miR-143/145 cluster described in DLBCL. An additional reason to think so is the fact that the most common hot-spots of mutations in the *TP53* gene in DLBCL are codons 175 and 248. The mutations in codons 135 and 273 also commonly occur in this disease (Fig. 1A) (*Bykov et al., 2018*; *Gurtner et al., 2016*; *Muller et al., 2014*). In studies on various malignancies, missense mutations in these codons have been correlated with lower activity in processing pri-miRNA to pre-miRNA. In particular, p53 is associated with Drosha/DGCR8 through interaction with p68 and p72. Mutant p.C135Y, p.R175H, p.R273H and p.R248Q p53 disrupt the assembly of the complex with Drosha by sequestering p68, while mutant p.R175H and p.R273H variants of p53 inhibit Drosha activity by direct binding proteins

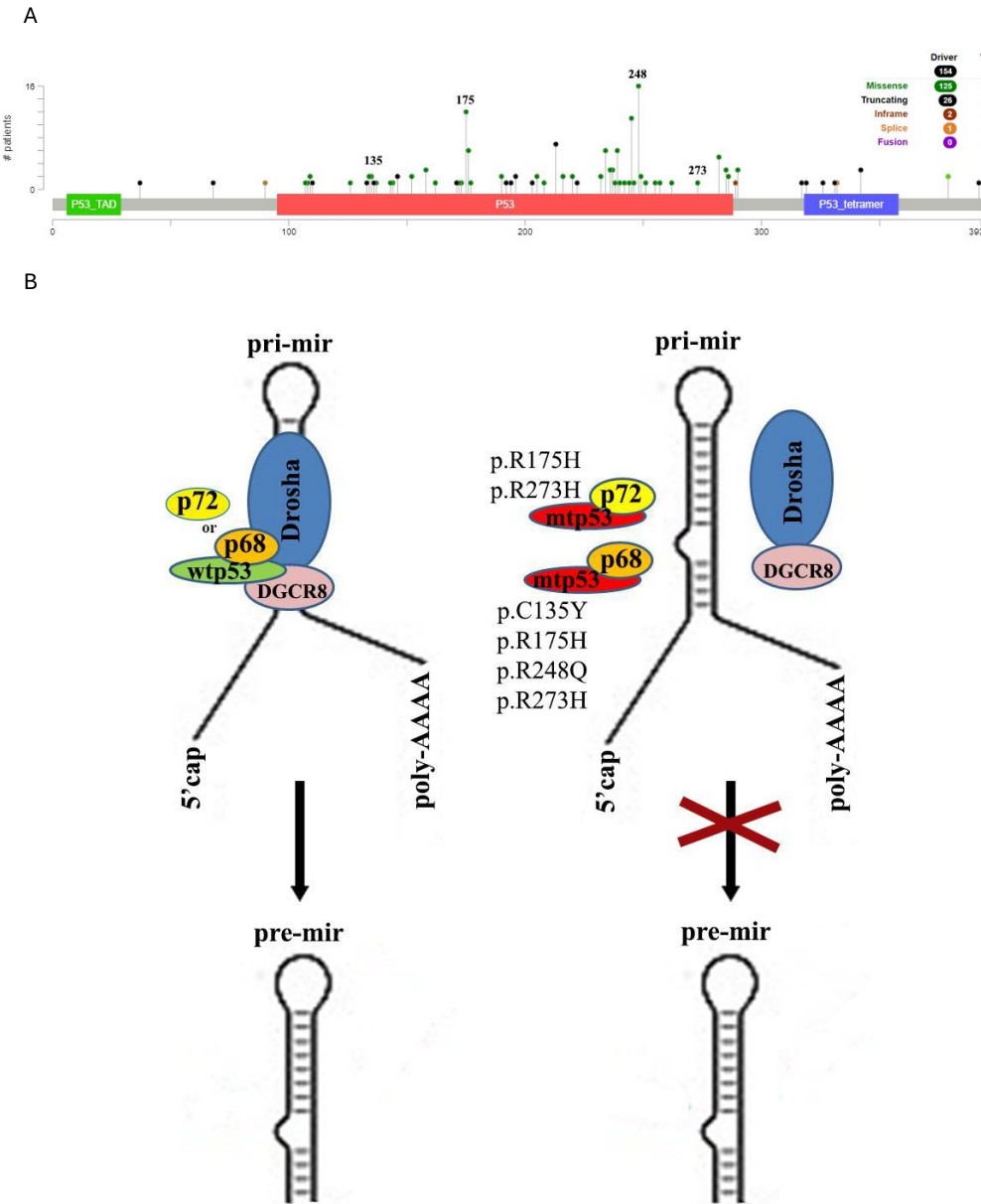

**Figure 1  Somatic mutations in the *TP53* gene in DLBCL.** (A) Distribution and characteristics of mutations in the *TP53* gene in DLBCL according to the results of next generation sequencing (NGS) analysis of 1,295 samples presented in the cBioPortal for Cancer Genomics database (*Gao et al., 2013*); (B) impaired maturation of pri-miRNA to pre-miRNA in the case of mutations in codons 135, 175, 248 and 273, which are mutation hotspots in the *TP53* gene.

p72 and p68 (Fig. 1B) (*Gurtner et al., 2016*). The observations of impaired processing of pre-miR to microRNA duplex under the influence of Dicer in the cytoplasm due to inhibition of p63 activity under the influence of p.R175H and p.R273H mutant p53 protein have been described (*Muller et al., 2014*).

In addition to mutations, of course, there are other mechanisms for dysfunction of the *TP53* gene. However, they are less common in DLBCL. There are a small number of published studies which purpose was to carry out assessment of other types of aberrations that potentially lead to impaired microRNA expression in DLBCL. For example, it is known that loss of 17p (locus of the *TP53* gene location) occurs in 16% (according to standard karyotyping) and *TP53* methylation—in 5–6% of cases at the time of diagnosis of DLBCL. Often these aberrations are combined in lymphoma tissue (*Voropaeva et al., 2019*; *Fiskvik et al., 2013*; *Voropaeva et al., 2023a*).

## The relationship between p53-responsive microRNA expression and host-gene expression

The relationship between p53-responsive microRNA expression and host-gene expression seems to be a less studied aspect. Transcription of microRNA genes is carried out by RNA polymerase (Pol II or Pol III). Intergenic miRNAs are transcribed independently using their own promoters, whereas intragenic miRNAs ones may be co-transcribed with the host gene. The dependence of miRNA expression on host gene expression and transcription factors has been demonstrated in several tumors. However, experimental evidence of such a correlation is still insufficient (*Kaller et al., 2022*).

The relationship between p53-dependent microRNAs expression and host gene expression also appears to be extremely complex. The genes *MIR-34A, MIR-34B/C, MIR-145* and *MIR-143,* unlike *MIR-129-1, MIR-129-2* and *MIR-203*, have an intragenic location, thus their expression potentially may depend on the expression host gene. However, at the present time there is no evidence of a relationship between the expression of the analyzed p53-responsive miRNA genes and their host genes in primary samples of DLBCL patients. This may be due to the following. All of the listed genes have been found to have their own promoters (*Zhang et al., 2022*; *Cavard et al., 2023*; *Strmsek & Kunej, 2014*; *Tang et al., 2016*). For example, for the genes of the miR-143/145 cluster, separate promoter regions and independent transcription factors have been identified (*Zeinali et al., 2019*; *Pidíkova, Reis & Herichova, 2020*). This is why, when *CARMN* is knocked down, the expression of miR-143 and miR-145 may decrease, but does not disappear completely (*Zhang et al., 2022*). Moreover, the transcriptional regulation of *MIR-34B/C* may be carried out not only from the promoter of the host gene, but also from the promoter of the oppositely oriented *BTG4* gene (*Cavard et al., 2023*).

## The significance of methylation of p53-responsive microRNA genes

The extensive analysis of genomic sequences of microRNA genes has shown that DNA methylation is one of the most common mechanisms for disrupting their transcription in malignant neoplasms (*Zeinali et al., 2019*). The microRNAs we describe are no exception. The promoters *MIR-34A, MIR-34B/C, MIR-129-2, MIR-203, MIR-143* and *MIR-145,* unlike *MIR-129-1* gene, also contain CpG islands, the aberrant methylation of which causes a decrease in microRNA expression in a wide range of neoplasms (*Voropaeva et al., 2023d*; *Voropaeva et al., 2022b*; *Voropaeva et al., 2023c*; *Benati et al., 2017*; *Wong et al., 2013*; *Gao et al., 2016*; *Hother et al., 2012*; *Chim et al., 2011*; *Voropaeva et al., 2023b*).

Methylation of microRNAs of the miR-34 family in DLBCL tumor tissue has been analyzed in several studies. According to the obtained results, the frequency of methylation in systemic DLBCL ranged from 23–28% for the *MIR-34A* gene and 55–78% for the *MIR-34B/C* gene (*Asmar et al., 2014*; *Voropaeva et al., 2022a*; *Wong et al., 2013*), while in DLBCL of central nervous system it was 57% and 95.2%, respectively (*Munch-Petersen et al., 2016*). According to literature methylation frequencies in the studied DLBCL samples were 65% for *MIR-129-2* and 66% for *MIR-203* (*Voropaeva et al., 2022a*). Moreover, comprehensive study of the gene methylation status in DLBCL samples showed that methylation of *MIR-34A, MIR-34B/C, MIR-129-2* and *MIR-203* in the tissue of affected lymph nodes of patients with DLBCL was a tumor-specific and combined phenomenon (*Voropaeva et al., 2023a*).

The works on quantitative assessment of the level of methylation of p53-responsive microRNA genes are of great interest. In particular, the data obtained by the Reduced Representation Bisulfite Sequencing method, which indicate an increase in the level of methylation of genes encoding miR-129 and miR-203 in relapsed DLBCL samples in comparison with diagnostic samples are very noteworthy and may signify an increasing role of methylation during tumor progression of lymphoma (*Leivonen et al., 2017*).

The above data indicate that methylation of microRNA genes in DLBCL appears to be a very promising therapeutic target. The reversibility of aberrant methylation of the promoters of the described genes was demonstrated in lymphoid tumor cell lines. Treatment of tumor cells with demethylating agents led to restoration of the level of mature microRNAs, inhibition of cell proliferation and triggering of cell death by apoptosis (*Wong et al., 2013*; *Chim et al., 2011*).

Thus, aberrant methylation may be one of the significant mechanisms of impaired expression of microRNAs miR-203, mir-129, miR-34a, miR-34b and miR-34c in DLBCL. At the same time, the methylation data for the miR-143/145 cluster are fundamentally different. Studies of *MIR-143* and *MIR-145* methylation in lymphoid tissue have shown that, on the one hand, the methylation of these genes occurs in reactive lymphadenopathy and thus is not tumor-specific feature in DLBCL. On the other hand, quantitative analysis has revealed a greater range of values of the average level of methylation of the *MIR-143* gene in samples from patients with DLBCL, but in general it was significantly lower than the values in samples from patients with reactive lymphadenitis (*Voropaeva et al., 2023c*; *Voropaeva et al., 2023b*). This, along with the inconsistency of data on the direction of changes in microRNA expression in DLBCL tumor tissue, indicates the need for further research into the mechanisms of epigenetic regulation of miR-143/145 cluster (*Akao et al., 2007*; *Lawrie et al., 2009*; *Voropaeva et al., 2023c*; *Voropaeva et al., 2023b*; *Roehle et al., 2008*; *Fischer et al., 2011*; *Vaisitti et al., 2018*).

## Copy number alteration in the loci of p53-responsive microRNA genes

According to recent data, up to half of all miRNA genes are located in genome regions that undergo amplification, deletion, or other structural rearrangements (*Zeinali et al., 2019*), and one of the possible reasons for the decreased expression of p53-responsive microRNAs in DLBCL may be the loss of entire chromosomes or deletion of chromosomal loci. For

example, the *MIR-129-1* gene is located in the fragile FRA7H site of chromosome 7 and is often lost in tumors (*Gao et al., 2016*), whereas recurrent chromosomal rearrangements involving the long arm of chromosome 14, where the *MIR-203* gene is located, and deletion of 11p11.2, the location of the *MIR-129-2* gene, are common in lymphomas including DLBCL (*Aya-Bonilla et al., 2011*; *Aya-Bonilla et al., 2013*; *Ricketts, Carter & Coleman, 2003*). It is shown that four protein coding genes oncosuppressors, including p53-induced protein (PIG11), also localized within 11p11.2-p12 (*Ricketts, Carter & Coleman, 2003*).

Interesting data from *Zhu et al. (2000)* showed that the deletion frequency of 11q23 (the location locus of *MIR-34B/C*) in a group of 17 DLBCL samples was very low (only 4.2%). At the same time, all three studied cases of transformation of indolent lymphoma into DLBCL, called the Richter syndrome, had a deletion of 11q23. This may indicate that gene loss is a late event in the evolution of lymphoma. In the work of *Hezaveh et al. (2016)*, whole-genome-derived copy number analysis did not reveal copy number violations in the promoter regions or regions of the miR-143/145 cluster genes in any of the 19 analyzed samples. As can be seen in the available literature, only a few studies of copy number disturbances of the microRNA genes in small sets of samples are described, which increases the practical significance of the analysis of available databases accumulating information on the molecular genetic characteristics of tumors.

It should be noted that karyotyping is widely used in hematology and is the main method used to search for and characterize chromosomal abnormalities in DLBCL (*Hezaveh et al., 2016*; *Wang & LaFramboise, 2019*). In this regard, we analyzed data from the Mitelman Database of Chromosome Aberrations and Gene Fusions in Cancer (*Mitelman Database of Chromosome Aberrations and Gene Fusions in Cancer, 2024*), which provides data on 1,612 cases of DLBCL with cytogenetic abnormalities. We discovered that the loss of entire chromosomes or deletions of their sections in the locations of the *MIR-34A, MIR-34B/C, MIR-129-1, MIR-129-2, MIR-203* genes and the miR-143/145 cluster occurred at 3.1%, 4.4%, 3.4%, 3.2%, 0.1% and 4.3%, respectively (Fig. 2A). In terms of all DLBCL cases including cases without cytogenetic aberrations these values may seem even less significant.

However, the classical cytogenetic method is known to have limitations. In particular, karyotyping does not allow detecting small chromosomal rearrangements that are outside the resolution range of the method (*Shah et al., 2017*). In this regard, the data presented in the cBioPortal for Cancer Genomics database are of great interest (*Gao et al., 2013*). The results of profiling of 396 DLBCL samples using Affymetrix SNP 6.0 microarrays were analyzed. It turned out that a decrease in genome copy number in the chr1:9,211,727-9,211,836 locus (GRCh37/hg19 by NCBI Gene), where the *MIR-34A* gene is located, occurred in 19% of cases. This value is lower than data by *Gru et al. (2013)*, who described a high incidence of microdeletion of the 1p36 region in non-Hodgkin lymphomas. According to their data, nine out of 32 (28%) studied DLBCL cases had 1p36 microdeletion by fluorescence *in situ* hybridization (FISH) technique, not recognized by conventional cytogenetics (*Gru et al., 2013*). Such differences in frequency may be primarily due to different sample sizes and composition, as well as methodological approaches.

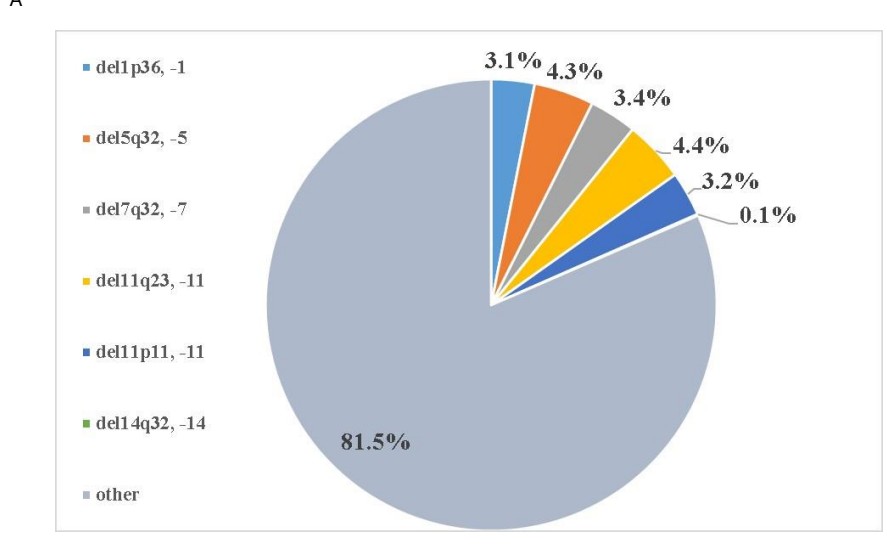

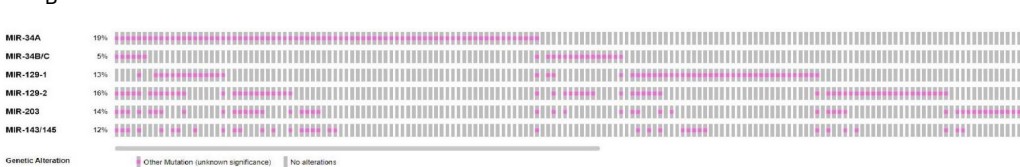

**Figure 2** **Disturbance of genome copy number in the loci of p53-responsive microRNA genes.** (A) The frequency of detected chromosomal abnormalities (monosomies and deletions of microRNA gene loci) according to a classic cytogenetic study of 1,612 DLBCL samples with cytogenetic defects presented in the Mitelman Database of Chromosome Aberrations and Gene Fusions in Cancer (*Mitelman Database of Chromosome Aberrations and Gene Fusions in Cancer, 2024*) (B) Cases with decrease in genome copy number in the loci of p53-responsive microRNA genes (rose color) presented in the cBioPortal for Cancer Genomics database (*Gao et al., 2013*).

Further, according to data, presented in the cBioPortal for Cancer Genomics database, the reduction of genome copy number in the gene loci *MIR-34B/C* (chr11:111,383,663-111,384,240), *MIR-129-1* (chr7:127,847,925-127,847,996), *MIR-129-2* (chr11:43,602,944-43,603,0 33), *MIR-203* (chr14:104,583,742-104,583,851), *MIR-143* and *MIR-145* genes (chr5:148,808,481-148,810,296), were observed in 5%, 13%, 16%, 14% and 12% of samples, respectively (Fig. 2B) (*Gao et al., 2013*; *Taylor et al., 2018*).

These data seem be extremely interesting. First, deletion or the loss of entire chromosome are the causes of *MIR-34B/C* gene loss, whereas the decrease in copy number of other described genes appears to be associated with microdeletions. Second, a decrease in copy number as a cause of decreased microRNA expression may have the greatest importance for miR-34a microRNA, and to a lesser extent for miR-129, miR-143, miR-145 and miR-203, whereas for miR-34b and miR-34c this mechanism may be insignificant. Third, as can be seen from Fig. 3B, the combined loss of two or more genes of p53-responsive microRNAs

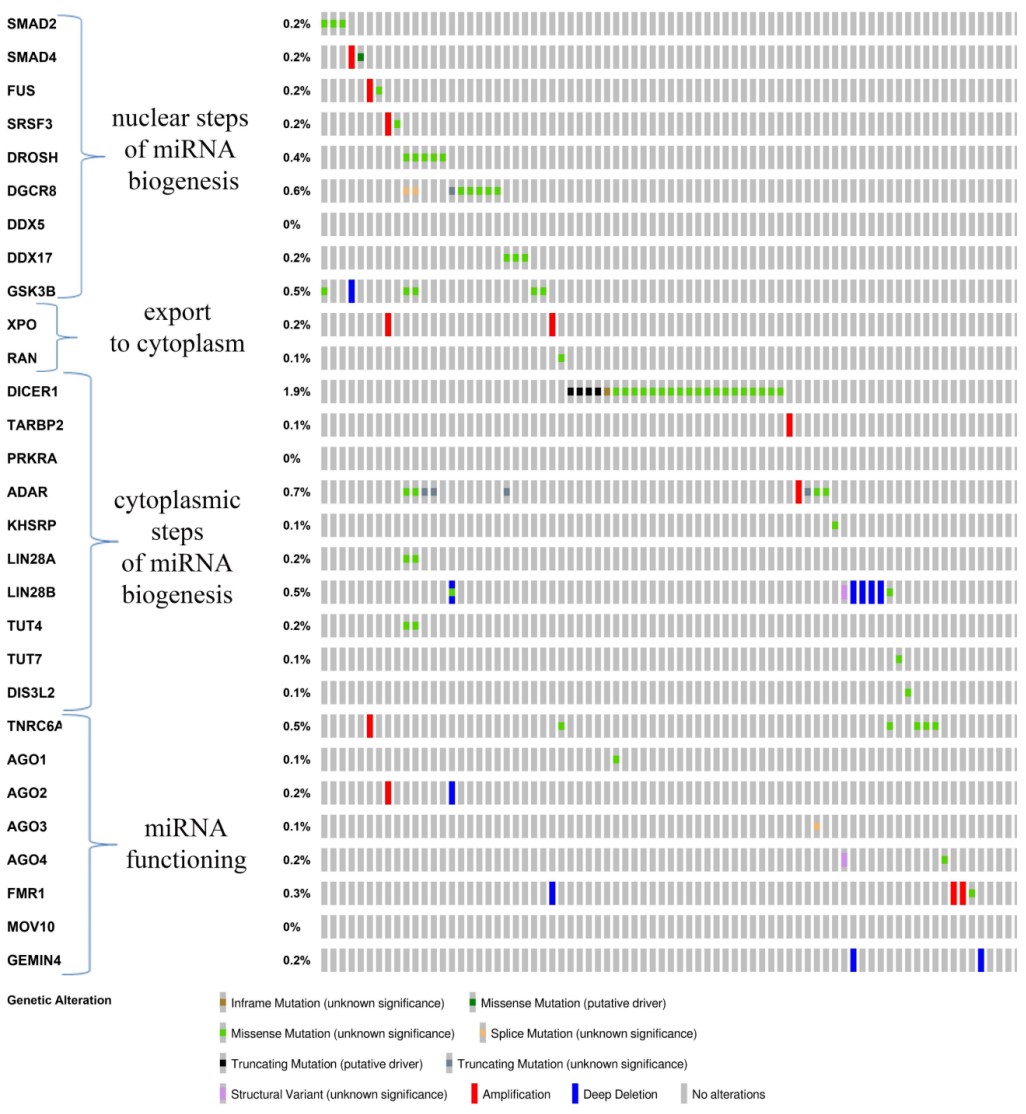

**Figure 3** Spectrum and frequency of detected aberrations in microRNA biogenesis genes according to the analysis of 1295 DLBCL samples presented in the cBioPortal for Cancer Genomics database (*Gao et al., 2013*).

often occurs in DLBCL. The combination of deletions is statistically significant even after correction for multiple comparisons for the gene pairs indicated in Table 2.

## Variants of the nucleotide sequence of p53-responsive microRNA genes

Until recently, very little attention was paid to changes in the nucleotide sequence of the genome outside the locations of protein-coding genes, because currently it is extremely difficult to assess their functional effect. The capabilities of bioinformatics for such an assessment are extremely limited. In this regard, the data on the frequency, structural and functional effect of somatic or germline variants of nucleotide sequence in miRNA genes are currently very limited (*Machowska, Galka-Marciniak & Kozlowski, 2022*). However,

**Table 2  The analysis of the tested pairs among the genes losses by the OncoPrinter, according to the cBioPortal for Cancer Genomics database (*Navarro & Lieberman, 2015*).**

| Gene 1 | Gene 2 | Log2 odds ratio | *p*-value | *q*-Value | Tendency |
|--------|--------|-----------------|-----------|-----------|----------|
| *MIR-129-2* | *MIR-203* | 2.669 | <0.001 | <0.001 | Co-occurrence |
| *MIR-129-2* | *MIR-34B/C* | >3 | <0.001 | <0.001 | Co-occurrence |
| *MIR-129-2* | *MIR-143/145* | 2.042 | <0.001 | <0.001 | Co-occurrence |
| *MIR-129-2* | *MIR-34A* | 1.766 | <0.001 | <0.001 | Co-occurrence |
| *MIR-143/145* | *MIR-34A* | 1.887 | <0.001 | <0.001 | Co-occurrence |
| *MIR-143/145* | *MIR-203* | 2.019 | <0.001 | <0.001 | Co-occurrence |
| *MIR-34A* | *MIR-203* | 1.577 | 0.001 | 0.003 | Co-occurrence |
| *MIR-129-1* | *MIR-143/145* | 1.548 | 0.005 | 0.010 | Co-occurrence |
| *MIR-129-1* | *MIR-129-2* | 1.347 | 0.012 | 0.021 | Co-occurrence |
| *MIR-34B/C* | *MIR-203* | 1.794 | 0.015 | 0.023 | Co-occurrence |

**Notes.**
The analysis of the combined microRNA genes losses was carried out using the one-sided Fisher's exact criterion (*p*-value) and multiple testing corrections with Benjamin–Hochberg procedure (*q*-value).

potential effects of mutations or single nucleotide polymorphisms in various parts of the miRNA gene sequence may be changes in the processes of transcription, biogenesis of miRNAs and their targets recognition. Thus, the decrease in miRNA expression may occur due to the mutations in the microRNA gene promoter or any part of the microRNA precursor sequence, including mutations in DROSHA/DICER1 cleavage sites or miRNA duplex, mutations disrupting/creating new PB motifs, *etc.* (*Machowska, Galka-Marciniak & Kozlowski, 2022*).

In the available literature, we have found only single descriptions of the spectrum of microRNA mutations that were detected at both the DNA and RNA levels in DLBCL samples. Using sequencing datasets from a large cancer genomics project, The Cancer Genome Atlas (TCGA), 36 miRNA mutations were described in 37 DLBCL samples. Most of them (92.7%) were found as substitutions, another 2.8% were nucleotide insertions. At the same time, 40.5% of the samples did not have any microRNA mutation. Among the mutated miRNA genes in DLBCL, the genes *MIR-34A, MIR-34B/C, MIR-129-1, MIR-129-2, MIR-203* and the miR-143/145 cluster are not included, unlike miR-1324 and miR-142 (*Urbanek-Trzeciak et al., 2020*; *Galka-Marciniak et al., 2021*). In one of the studies in a set of 19 samples of DLBCL, the authors reported the detection of mutations in the genes of only three microRNAs, namely miR-142, miR-612 and miR-4322 (*Hezaveh et al., 2016*). In *Kwanhian et al. (2012)*, no other mutations except in miR-142 were observed in the analysis of 56 lymphoma samples (*Kwanhian et al., 2012*).

These data suggest that nucleotide sequence variants in the p53-responsive miRNA genes we analyzed are at least a rare event in DLBCL. However, it is undoubtedly necessary to conduct studies with a large sequencing depth and a large number of DLBCL samples included as well as to develop new methods of statistical and bioinformatics processing in order to clarify the frequency and improve understanding of the effects of genetic variants in microRNA genes.

## Mutations in microRNA biogenesis genes

The regulation of miRNA biogenesis is complex, tightly controlled, and may be disrupted by various factors such as mutations or epigenetic modifications, and the expression level is the outcome of the transcription and subsequent biogenesis of microRNAs, in which a large number of proteins are involved (*Gulyaeva & Kushlinskiy, 2016*). All microRNA biogenesis genes are participants of several steps, namely nuclear step of miRNA biogenesis (*SMAD2/4, FUS, SRSF3, DROSHA, DGCR8, DDX5/17* and *GSK3B*), export to cytoplasm (*XPO5* and *RAN*), cytoplasmic step of miRNA biogenesis (*DICER1, TARBP2, PRKRA, ADAR, KHSRP, LIN28A/B, TUT4/7, DIS3L2*) and miRNA functioning (*TNRC6A, AGO1/2/3/4, FMR1, MOV10, GEMIN4*) (*Galka-Marciniak et al., 2021*; *Annese et al., 2020*). It has been reported that mutation frequencies in some of these genes specifically and significantly increased in certain types of cancer. However, none of the 29 microRNA biogenesis genes could be classified as over mutated genes in DLBCL (*Jafari et al., 2015*).

Analysis of whole exome sequencing data sets of 37 paired tumor and normal samples from patients with DLBCL, available in the TCGA repository, has shown the presence of mutations in only seven (*TNRC6A, MOV10, DGCR8, DIS3L2, GEMIN4, LIN28B* and *RAN*) out of 29 microRNA biogenesis genes. In total, 16% of samples have had mutations in microRNA biogenesis genes (*Galka-Marciniak et al., 2021*).

According to the analysis of high-throughput sequencing data of 1295 DLBCL samples presented in the cBioPortal for Cancer Genomics database, aberrations rarely detected in all genes except for three, namely *MOV10, PRKRA* and *DDX5* (*Mitelman Database of Chromosome Aberrations and Gene Fusions in Cancer, 2024*). A total of 6.9% of samples had aberrations in microRNA biogenesis genes, while disorders that are presumably drivers of the tumor process (missense and truncating mutations, deep deletions) were identified in a small number of cases (Fig. 3). However, two samples of DLBCL are noteworthy, in which a significant combination of aberrations was identified in six functionally related genes of the nuclear (*DROSHA, DGCR8* and *GSK3B*) and cytoplasmic (*TUT4, LIN28* and *ADAR*) stages of microRNA biogenesis.

The function of the affected in these two samples of DLBCL genes is briefly described below. Drosha and DGCR8 are considered as major machinery components of miRNA biogenesis nuclear steps (*Jafari et al., 2015*). The *GSK3B* gene encodes the GSK3 $\beta$ kinase, which has been shown recently to bind to DGCR8 and p72 in the microprocessor and thus stimulate the Drosha-cofactor and Drosha-pri-miR interaction (*Fletcher et al., 2017*).

The Lin28a protein, whose biochemical activity is indistinguishable from that of the homologue Lin28b, is a regulatory factor in the cytoplasmic stage of microRNA biogenesis. Lin28 binds to pre-microRNA after export from the nucleus to the cytoplasm. Subsequently, another protein (TUT4 protein) recognizes the Lin28-pre-miRNA complex and performs its uridylation, which makes the pre-miRNA resistant to Dicer processing. It has also been reported that Lin28 proteins interfere with the processing of pri-miRNA to pre-miRNA by Drosha (*Heo et al., 2009*). Abnormalities in the *TUT4* and *LIN28* genes, encoding specific suppressors of miRNA biogenesis, are involved in carcinogenesis.

The adenosine deaminase acting on RNA (ADA) enzyme converts adenosine to inosine in double-stranded RNAs, including microRNAs duplex, which leads to disruption of

miRNA maturation process and activity (*Tomaselli et al., 2013*). *ADAR* knockout can cause hematological tumors in animal models (*Correia De Sousa et al., 2019*).

The mutations found in these genes are regarded as variants of unknown significance, but their potential effect may be a disruption of the basic biogenesis of microRNAs, and an additional study is required to prove it.

## CONCLUSION AND FUTURE PERSPECTIVE

The research focused on microarray expression profiling of miRNAs shows that general microRNA deregulation is a common event in neoplasms: tumors have their own spectrum of microRNAs that is different from normal cells (*Jones & Lal, 2012*). In this regard microRNAs are recognized markers with diagnostic and prognostic values, as well as potential targets for the treatment of a wide range of tumors. At the same time, at the present stage of scientific development, there is still a lack of full understanding of how the expression of each of the microRNAs in normal cells is controlled in general, and what underlies the disruption of its expression in various diseases, including malignant neoplasms. It is known that one of the main characteristics of tumor cells is genetic instability, manifested by the accumulation of damage at the different levels of the organization of hereditary material: chromosomal aberrations, microsatellite instability, gene mutations, as well as epigenetic changes (disturbances in methylation patterns, histone modifications), *etc.* (*Garnett & McDermott, 2012*), which can lead to disruption of microRNA expression deregulation at various levels. Arising from this inter-patient and intra-tumor heterogeneity present in reality will inevitably prevent the transfer of therapeutic success of correction of one or another cause of microRNA deregulation achieved in experiments in cell lines to real patients.

Aberrant expression of microRNAs is also observed in lymphomas, including DLBCL—the most frequent and aggressive variant of non-Hodgkin's lymphomas. Not only differences in the level of microRNAs in lymphoma cells and normal lymphoid tissue have been shown, but also the possibility of dividing DLBCL into prognostically different subgroups of the disease from cells of germinal origin and activated B cells, based on analysis of the spectrum of expressed microRNAs (*Larrabeiti-Etxebarria et al., 2019*; *Larrabeiti-Etxebarria et al., 2023*; *Lawrie et al., 2009*).

We are interested in some p53-responsive microRNAs miR-34a, miR-34b, miR-34c, miR-129, miR-203, miR-145 and miR-143, which play tumor suppressor role in DLBCL. Several studies have demonstrated a decrease in the expression of these microRNAs in lymphoma tissue (*Hedström et al., 2013*; *Akao et al., 2007*; *Isaadi et al., 2021*; *Yamagishi et al., 2015*; *Zheng et al., 2021*; *Larrabeiti-Etxebarria et al., 2023*; *Mazan-Mamczarz & Gartenhaus, 2013*). Moreover, low miR-129 expression was found to be associated with shorter survival in DLBCL patients both with and without R-CHOP treatment, containing a combination of chemotherapy drugs cyclophosphamide, doxorubicin hydrochloride (hydroxydaunomycin), and vincristine sulfate (oncovin), the targeted therapy drug rituximab, and the steroid hormone prednisone (*Hedström et al., 2013*). Low miR-34a level was also found to be associated with chemoresistance (*Larrabeiti-Etxebarria et al., 2019*).

Taking into account the fragmentation of available data, we reviewed a number of potential molecular mechanisms in the disruption of these microRNAs expression in DLBCL. The data presented in this review provide a more comprehensive, but far from complete, understanding of the possible reasons for the decrease in the level of p53-responsive microRNAs miR-34a, miR-34b, miR-34c, miR-129, miR-203, miR-145 and miR-143 in DLBCL.

We can conclude that the disruption of microRNA biogenesis resulting from variants in the nucleotide sequence of the genes encoding them or variants in the nucleotide sequence of the genes of key proteins involved in miRNA biogenesis does not seem to have a significant importance in DLBCL. On the contrary, changes in the number of somatic gene copies may be significant for impaired expression of described p53-responsive microRNAs in DLBCL except for miR-34b. For example, the *MIR-34A* gene is lost, mainly through the mechanism of microdeletions in every fifth case, and the *MIR-129-1, MIR-129-2, MIR-203* and miR-143/145 cluster genes are lost in DLBCL somewhat less frequently.

Also the aberrant methylation of the regulatory sequences of the genes *MIR-34A, MIR-34B/C, MIR-129-1, MIR-129-2* and *MIR-203*, but not *MIR-145* and *MIR-143*, is the most common, tumor specific and combined phenomenon among various mechanisms of deregulation of described microRNAs expression in DLBCL. In recent years, liquid biopsy has become increasingly important in oncology. Its capabilities are being studied not only for the purpose of early diagnosis of tumors, but also to assess the depth of response to treatment, monitor the level of minimal residual disease and early diagnosis of disease recurrence (*Talotta et al., 2023*). As stated above, miRNAs genes methylation associated with miRNAs deregulation in tumor lymphoid tissue can be utilized as diagnostic biomarker for DLBCL. Assessment of the methylation status of target microRNA genes by free circulating tumor DNA requires further study and could become one of approaches to liquid biopsy in patients with DLBCL in the future. The technical possibility of assessing the status of DNA methylation in the framework of liquid biopsy in DLBCL was shown earlier (*Wedge et al., 2017*).

Aberrant methylation of studied microRNAs genes has therapeutic implication, because hypomethylating agents may be useful to eliminate such methylation. The reversibility of methylation of the promoter *MIR-203* and *MIR-129-2* when treated with 5-Aza-2′-deoxycytidine was shown on Hodgkin's lymphoma and non-Hodgkin's lymphoma cell lines, which led to the restoration of expression of microRNAs encoded by these genes, inhibition of cell proliferation or induction of tumor cell death (*Peng et al., 2020*; *Chim et al., 2011*; *Xu et al., 2022*; *Fatema, Larson & Barrott, 2022*). Such hypomethylating agents as azacitidine and decitabine have already been approved for the treatment of blood malignancies, for example, myelodysplastic syndrome and acute myeloid leukemia, associated with global genome hypermethylation phenomena (*Zhou et al., 2020*). However, recent studies have shown that the global methylation level in DLBCL is characterized by high variability (*Chambwe et al., 2014*). In particular, there are cases of lymphoma with global genome hypomethylation (*Wedge et al., 2017*). These data indicate the need to select patients with DLBCL for therapy with hypomethylating agents. Moreover, due to the nonspecific nature of their action, simultaneously with a decrease in the level of

methylation of CpG islands in the promoters of oncosuppressor genes, hypomethylating agents can potentially cause a decrease in the level of methylation of proto-oncogenes, which can contribute to the progression of the disease (*Jun et al., 2019*). Ongoing clinical trials in combined hypomethylating agents therapy (high-dose chemotherapy or chimeric antigen receptor (CAR) T-cell immunotherapy) with other treatments aim to determine the optimal therapy dose and synergy effect in the treatment of in refractory/relapsed DLBCL patients (*Kalinkova et al., 2022*).

Moreover, it can be assumed that the removal of aberrant methylation itself may not lead to the desired effect in the case of combined disorders in the tumor that lead to disruption of microRNA expression. Unfortunately, with rare exceptions, we were unable to find information on a comprehensive analysis of several mechanisms of microRNA expression disruption in DLBCL samples. From the available data it should be mentioned the study of methylation of the *MIR-34A, MIR-34B/C, MIR-129-2* and *MIR-203* genes and aberrations (mutations and destruction of the polyadenylation signal) in the *TP53* gene in DLBCL which has shown that in the tissue of the affected lymph nodes of DLBCL patients these violations were independent (*Fiskvik et al., 2013*).

As shown in the previous section of the review mutations in the *TP53* gene are verified in 20% and more of DLBCL cases. Various genomic editing options including base editing, prime editing and upcoming technologies, are applicable to correct mutations. Although there has been an increase in the number of studies revealing the possibilities of CRISPR/Cas9 (clustered regularly interspaced short palindromic repeats/Cas9) technology for editing the *TP53* gene, such approach seems practically feasible in the extremely distant future (*Mirgayazova et al., 2020*).

Dysfunction of p53 as a cause of deregulation of p53-responsive microRNAs requires further research, because the following data add additional complexity to the current picture. Transcription of p53-responsive microRNAs may be regulated not only by p53, but also by other transcription factors (*Kaller et al., 2022*; *Asmar et al., 2014*). The level of microRNAs in cells changes through a feedback loop depending on the level of expression of target mRNAs and is regulated by various modifications affecting the stability of these molecules, for example, adenylation (*Kaller et al., 2022*). However, these factors have not been studied at all in DLBCL and may vary depending on the type of cells and the level of cell differentiation.

Resolving these issues and deepening the understanding of aberrant expression of microRNAs in tumors, including DLBCL, can be achieved through comprehensive studies of various causes of their deregulation using a single large set of tumor biospecimen.

Regardless of the cause of the microRNA expression disorder, a universal therapeutic approach in DLBCL could potentially be microRNA mimics. Some such drugs including MRX34 (miR-34 mimic in a liposome) are undergoing preclinical and clinical trials (*Winkle et al., 2021b*; *Hong et al., 2020*; *Chakraborty et al., 2020*). Thus, the effects of miR-34 restoration in cancer cells lines by transfected with miR-34 mimics or infected with the lentiviral miR-34 expression system were as follows: reduced the expression of target oncogenes (Bcl-2, Notch *etc.*), chemosensitized, impaired cell growth, accumulated the cells in G1 phase, increased apoptosis (*Ji et al., 2008*). However, the results of such studies

in human are still not encouraging due to large number of immune-related side effects up to fatal outcomes and serious adverse events associated with miRNA overexpression demanding data integration (*Mirgayazova et al., 2020*; *Golebiewski et al., 2024*). Obviously, further advances in this field are possible provided that more natural analogues, safe and effective strategies for targeted delivery of these therapeutic agents to the cells of interest are developed, ensuring the specificity of their action (*Huang et al., 2023*).

## ACKNOWLEDGEMENTS

The authors are grateful to the reviewers for the positive comments and novel reference suggestions.

### Funding

The study p53-responsive microRNA was supported by the topic of government assignment (Project No. FWNR-2024-0004) (E.V. and colleagues). Bioinformatics analysis was supported by Russian Science Foundation (project 24-24-00563) (Yuriy L. Orlov). The funders had no role in study design, data collection and analysis, decision to publish, or preparation of the manuscript.

### Grant Disclosures

The following grant information was disclosed by the authors:
Topic of Government Assignment: FWNR-2024-0004.
Russian Science Foundation: 24-24-00563.

### Competing Interests

Yuriy L. Orlov is an Academic Editor for PeerJ.

### Author Contributions

- Elena N. Voropaeva conceived and designed the experiments, performed the experiments, analyzed the data, prepared figures and/or tables, authored or reviewed drafts of the article, and approved the final draft.
- Yuriy L. Orlov conceived and designed the experiments, performed the experiments, authored or reviewed drafts of the article, and approved the final draft.
- Anastasia B. Loginova performed the experiments, prepared figures and/or tables, and approved the final draft.
- Olga B. Seregina conceived and designed the experiments, authored or reviewed drafts of the article, and approved the final draft.
- Vladimir N. Maksimov analyzed the data, authored or reviewed drafts of the article, and approved the final draft.
- Tatiana I. Pospelova conceived and designed the experiments, prepared figures and/or tables, and approved the final draft.

## Data Availability

This is a literature review.

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
