# Peer review of "Deregulation mechanisms and therapeutic opportunities of p53-responsive microRNAs in diffuse large B-cell lymphoma"

_PeerJ, doi:10.7717/peerj.18661_

## Round 0.1 · original submission · Minor Revisions

The reviewers have called for some minor revisions, please read their comments carefully and provide an updated MS when possible.

Reviewer 1 ·

Basic reporting

This review concerned the molecular mechanisms of interested microRNA dysregulation in response to genotoxic stress in Diffuse Large B-cell Lymphoma (DLBCL) patients. The dield has not been reviewed recently.
The authors effectively introduce the subject by outlining the significance of microRNAs in oncogenesis and the potential implications for cancer therapy. Overall, the Introduction is adequate in setting the stage for the research and clearly identifies the intended audience and motivation.

Experimental design

Article content is within the Aims and Scope of the journal and article type. The Methods section provides a detailed account of the literature search strategy, including the databases used, specific keywords, and the time frame of the publications analyzed. Additionally, the section specifies the databases and tools used for data on microRNA-target interactions, karyotyping, and genetic data, such as miRTarBase. The sources and tools are well-documented. Overall, the methods are adequately described but could benefit from additional clarity to ensure full replicability. Sources were adequately cited. The review was organized logically into coherent paragraphs.

Validity of the findings

The conclusions presented are well-aligned with the original research question, focusing on the role and regulation of p53-responsive microRNAs in DLBCL. The authors effectively summarize the key findings, particularly the mechanisms of microRNA deregulation, such as changes in gene copy number and aberrant methylation.
Also, the conclusion effectively highlights unresolved questions and gaps in the current understanding of microRNA deregulation in DLBCL. The authors point out the need for further research into the comprehensive analysis of multiple mechanisms of microRNA expression disruption, the potential for CRISPR/Cas9 editing of TP53 mutations, and the complexities of p53-responsive microRNA regulation by factors other than p53.
While the conclusions section was comprehensive, it could benefit from greater clarity and focus. Some of the key findings are presented in a way that may overwhelm the reader. In addition, the review would be beneficial to provide more detailed examples of future research directions, particularly regarding the mechanisms of microRNA regulation and the development of therapeutic strategies. For instance, suggesting specific experimental approaches or technologies that could be explored would strengthen this section.

Reviewer 2 ·

Basic reporting

The English language in the complex is clear enough, but there are few periods that are too long or complicated and should be rephrase in order to make text easier for the international audience to understand. Some examples where language should be revised are: the first sentence of introduction in line 48; line 70; 119; 272 . Despite this note, the text as a whole is clear.

Experimental design

no comment

Validity of the findings

no comment

Additional comments

no comment

Annotated reviews are not available for download in order to protect the identity of reviewers who chose to remain anonymous.

·

Basic reporting

In the present manuscript authors addressed the dysregulation molecular mechanisms of p53-mediated microRNAs that open a novel therapeutic opportunity in management of diffuse large B-cell lymphoma (DLBCL). The role of p53 in microRNA regulation in numerous cancers including DLBCL has been reported by various researchers. As microRNAs are crucial regulators of various cellular processes like proliferation, differentiation and apoptosis, hence it might be better therapeutic approach to targeting the deregulated p53/miRNA axis in treatment of DLBCL. Therefore, the idea of manuscript seems novel and deals with dysregulation of p53-resposnive microRNA in DLBCL. Authors have written the manuscript in a lucid manner.
BASIC REPORTING:
-Clear, unambiguous, professional English language used throughout- Yes
But some grammatical errors are found in the manuscript which should be corrected:
Line No.28- It should be “Here, we have discussed the” instead of we discuss
Line No.31: it should be “at post-transcriptional level” instead of on post-transcriptional level
Line No.28: Please use some most appropriate word instead of interested microRNA
Line No.54: it should be “at post-transcriptional level” instead of on post-transcriptional level
Line No.65: Please deleted “of” word from the understanding of the effect
Line No.97: Please insert “have” word in we have analyzed
Line No.188: Please put comma after also word
Line No.235: Please correct the spelling eexperimental. Write the “experimental” instead of eexperimental
Line No.300: Please remove extra in word

-Intro & background to show context-Yes
-Literature well referenced & relevant Structure conforms to PeerJ standards, discipline norm, or improved for clarity.
Updated literature is cited.
-Is the review of broad and cross-disciplinary interest and within the scope of the journal?
Yes
-Has field been reviewed recently? It there a good reason for this review (different viewpoint, audience etc.)?
Not exactly but a recent review has addressed the importance of p53 gene in DLBCL that is relevant to poor prognosis. The better understanding of abnormalities of p53 is meant for the basis of development of better therapeutic strategy for DLBCL (Wen et.al., 2024).
Wen W, Zhang WL, Tan R, Zhong TT, Zhang MR, Fang XS. Progress in deciphering the role of p53 in diffuse large B-cell lymphoma: mechanisms and therapeutic targets. Am J Cancer Res. 2024 Jul 15;14(7):3280-3293. doi: 10.62347/LHIO8294.
Authors have mentioned the importance of p53-miRNA network in diagnostic and therapeutic approaches for numerous cancers (Sargolzaei et. al., 2020).
Sargolzaei J, Etemadi T, Alyasin A. The P53/microRNA network: A potential tumor suppressor with a role in anticancer therapy. Pharmacol Res. 2020 Oct;160:105179. doi: 10.1016/j.phrs.2020.105179. Epub 2020 Sep 3. PMID: 32890739.

-Introduction adequately introduces the subject and makes audience and motivation clear.
Introduction displays the enough evidences to support subject matter of manuscript.

Experimental design

STUDY DESIGN:
-Article content is within the Aims and Scope of the journal.
Yes
-Rigorous investigation performed to a high technical & ethical standard.
N.A
-Methods described with sufficient detail & information to replicate.
N.A
-Is the Survey Methodology consistent with a comprehensive, unbiased coverage of the subject? If not, what is missing?
Yes
-Are sources adequately cited? Quoted or paraphrased as appropriate?
Yes
-Is the review organized logically into coherent paragraphs/subsections?
Yes

Validity of the findings

VALIDITY OF THE FINDINGS
-Impact and novelty is not assessed. Meaningful replication encouraged where rationale & benefit to literature is clearly stated.
-Conclusions are well stated, linked to original research question & limited to supporting results.
Yes
-Is there a well developed and supported argument that meets the goals set out in the Introduction?
The figures drawn are clear and illustrate the results in finer details. The figure legends have been written in well interpretable manner. The Tables drawn are neat and help in better understanding of manuscript.
-Does the Conclusion identify unresolved questions / gaps / future directions? Yes
Although they have very well written conclusion of the present manuscript but it could be better if they will also write future prospective.

Additional comments

Miscellaneous Comments:
-Abbreviations should be mentioned in the manuscript.
-What is the role of mutant p53 in regulation of miRNA in carcinogenesis?
-Authors need to write the future prospects of the manuscript in separate section after conclusion?
Although, manuscript is very well written and idea is novel but minor revision is required in the manuscript. Henceforth, I will recommend the manuscript for publication after incorporation above changes in the manuscript.

---

## Round 0.2 · accepted · Accept

Dear authors,

I assessed your revision and am happy that you addressed the reviewers comments. The article is ready for publication.